# High Prevalence of Sticky Platelet Syndrome in Patients with Infertility and Pregnancy Loss

**DOI:** 10.3390/jcm8091328

**Published:** 2019-08-28

**Authors:** Eray Yagmur, Eva Bast, Anja Susanne Mühlfeld, Alexander Koch, Ralf Weiskirchen, Frank Tacke, Joseph Neulen

**Affiliations:** 1Medical Care Center, Dr. Stein and Colleagues, D-41169 Mönchengladbach, Germany; 2Department of Gynecology and Obstetrics, Bürgerhospital Frankfurt, D-60318 Frankfurt, Germany; 3Division of Nephrology and Clinical Immunology, RWTH-University Hospital Aachen, D-52074 Aachen, Germany; 4Department of Medicine III, RWTH-University Hospital Aachen, D-52074 Aachen, Germany; 5Institute of Molecular Pathobiochemistry, Experimental Gene Therapy and Clinical Chemistry, RWTH-University Hospital Aachen, D-52074 Aachen, Germany; 6Department of Hepatology and Gastroenterology, Charité University Medical Center, D-10117 Berlin, Germany; 7Department of Gynecological Endocrinology and Reproductive Medicine, RWTH-University Hospital Aachen, D-52074 Aachen, Germany

**Keywords:** platelets, hyperaggregability, sticky platelet syndrome, platelet function, infertility, miscarriage, pregnancy loss

## Abstract

Platelet hyperaggregability, known as sticky platelet syndrome (SPS), is a prothrombotic disorder that has been increasingly associated with pregnancy loss. In this retrospective study, we aimed to investigate the clinical and diagnostic relevance of SPS in 208 patients with infertility and unexplained pregnancy loss history. We studied 208 patients that had been referred to undergo a dose-dependent platelet aggregation response to adenosine diphosphate and epinephrine using light transmission aggregometry modified by Mammen during an 11-year period. Patients’ platelet aggregation response was compared with platelet function in 29 female healthy controls of fertile age with no previous history of pregnancy loss. We found a prevalence of SPS type II (33.2%) in 208 female patients with infertility and pregnancy loss. ∆-epinephrine-induced platelet aggregation in patients with SPS was significantly decreased (median 7% and range −21 to 43%) compared to patients without SPS (median 59%, range 7–88% and *p* < 0.0001) and healthy controls (median 57%, range 8–106% and *p* < 0.0001). The optimum SPS-diagnostic cutoff value for ∆-epinephrine aggregation was ≤32% (sensitivity 95.7%, specificity 95.2%). SPS patients with low-dose acetylsalicylic acid (ASA) therapy (*n* = 56) showed improved pregnancy outcome (32 pregnancies; live births *n* = 18 (56%)) compared to SPS patients without low-dose ASA (*n* = 13) (3 pregnancies; live births *n* = 1 (33%)). Our study demonstrates the clinical and diagnostic relevance of platelet hyperaggregation in women with infertility and pregnancy loss history. Further studies should investigate the potential of SPS as a novel decisional tool with both diagnostic and clinical implications in infertility and pregnancy loss.

## 1. Introduction

An unfulfilled desire to have children is a serious medical and social phenomenon. More and more women are being treated for infertility and sterility. Approximately 10–15% of couples remain childless despite a persistent desire to have children and 1–3% of all couples experience recurrent miscarriages [1,2]. For the woman who is affected, this problem is a clear limitation on the quality of life, and a profound dilemma for the partnership [3]. The diagnosis and therapy of couples with an unfulfilled desire to have children clearly confront the health professionals treating them with challenges, since although some causes are known, the majority of them remain undiscovered [4]. However, infertility and pregnancy loss are common gynecological or reproductive problems caused by well-defined deficiencies associated with chromosomal, anatomic and hormonal abnormalities (progesterone, estrogens, diabetes, thyroid disease) [5,6]. The American Society of Reproductive Medicine (ASRM) defines infertility as failure to conceive after 1 year of appropriately timed unprotected intercourse. Further common causes of infertility are pelvic diseases (tubal disease, fibroids, endometriosis), male factor, disorders of ovulation, female reproductive aging and idiopathic causes. Infertility and pregnancy loss can be interdependent, often resulting in a combined disorder [4]. In this context, the thrombotic hemostasis disorders associated with pregnancy loss include lupus anticoagulants and anticardiolipin antibodies, factor XII deficiency, dysfibrinogenemias associated with thrombosis, protein C defects, protein S defects, antithrombin deficiency, heparin cofactor II deficiency and fibrinolytic defects associated with thrombosis [1,7,8,9,10,11,12,13,14]. Notably, pregnancy itself is a hypercoagulable state predisposing to uteroplacental thrombosis, owing at least in part to the changes in coagulation factors, their regulators and fibrinolytic systems [6,15]. In addition, low pressure and turbulent flow of the placental perfusion may also predispose to thrombosis [16]. Thus, impaired placental circulation and hypercoagulability are involved in gynecological or reproductive complications such as recurrent pregnancy loss that occurs in 1–5% of pregnant women [2,6]. While platelets have very important roles in hemostasis [17], excess activation of platelets or platelet hyperaggregability is involved in the pathophysiological processes underlying pregnancy loss or unexplained thrombosis, which have also been confirmed in diverse animal models [6,18,19,20,21]. Thus, although not often screened for or recognized, perturbations of platelet function can be an etiology for prothrombotic conditions in pregnancy [15]. As a consequence of platelet hyperaggregability, acetylsalicylic acid (ASA) reduces platelet aggregation by inhibiting the enzymes cyclooxygenase-1 (COX-1) and cyclooxygenase-2 (COX-2), leading to the reduced formation of thromboxane A2 (TXA2), a potent stimulator of platelet activation, thereby reducing platelet secretion and aggregation [22].

Al-Mefty et al. first described platelet hyperaggregability in a group of young adults with unexplained repetitive transient ischemic attacks who were noted to have hyperaggregable platelets when exposed to adenosine diphosphate and epinephrine [23]. Nowadays, platelet function, particularly platelet hyperaggregability, is characterized by light transmission aggregometry and dose-independent platelet aggregation solely due to epinephrine and/or adenosine diphosphate (ADP) [24]. In the early 1980s, Mammen and Bick suggested platelet hyperaggregation, or SPS, as the underlying cause of thrombophilia in patients with unexplained arterial and venous thromboembolism or recurrent spontaneous abortions [6,18]. They named platelet hyperaggregability, in response to either epinephrine or adenosine or both sticky platelet syndrome (SPS). This laboratory phenomenon has later been clinically described in children and young adults with no identifiable risk factors for thrombosis and an otherwise negative thrombophilia evaluation [18]. Furthermore, SPS has been described by numerous clinical case reports associated with acute myocardial infarction, chronic kidney disease and thromboembolic kidney graft infarction [6,18,24]. Further studies have revealed SPS as the second most common hereditary thrombophilia after resistance to activated protein C and the most common thrombophilia associated with arterial thrombosis, with a reported incidence of approximately 21% in unselected populations [18,25]. Although the incidence of SPS in pregnancy loss is not well known, it has been described in women with recurrent pregnancy loss without a history of thrombosis [15]. Such cases are presently limited to small descriptive observations [15,24]. Therefore, SPS may be an underappreciated etiology for infertility and pregnancy loss [15]. 

Although SPS is the second most common thrombophilia that causes recurrent spontaneous abortions or fetal loss syndrome [15,26], this potential association has not yet been investigated epidemiologically in the field of miscarriage diagnostics. In previous investigations of the causal associations of platelet hyperaggregability, miscarriages and SPS have frequently been reported together, but no investigation has so far looked more closely at the pregnancy outcome in patients with SPS in addition to prevalence analyses. Based on these considerations, we aimed to investigate retrospectively the platelet aggregability in a large patient cohort with infertility and pregnancy loss history, in comparison to healthy non-pregnant women of fertile age. In this explorative study, we evaluated the thrombocyte function diagnostics modified according to Mammen with regard to the following aspects: (a) investigation of the prevalence of SPS, (b) diagnostic value of thrombocyte function analysis modified according to Mammen, and (c) retrospective assessment of the pregnancy outcome with and without ASA therapy with respect to the presence of SPS.

## 2. Experimental Section

### 2.1. Study Design and Patient Characteristics

The study included 208 female patients, who were admitted to the Department for Gynecological Endocrinology and Reproductive Medicine at the RWTH Aachen University Hospital between February 2001 and January 2012. All patients were presented for potential in vitro fertilization (IVF) due to infertility evaluation (duration of infertility >1 year) or diagnostic assessment of unexplained pregnancy loss history. All consecutive patients received platelet aggregation analysis. In order to compare the platelet aggregability analysis performed during this period, we conducted platelet function tests in 29 age-matched healthy, non-pregnant, fertile women as a diagnostic control group (Table 1). The two groups, patients and healthy women, were 20–49 years old at the time of testing the platelet function analysis (patients: median 34 years; healthy women: median 36 years; *p* = 0.1988, not significant). These healthy controls were investigated to determine the decisional platelet hyperaggregability cut-offs. Platelet hyperaggregability in patients was assessed by comparison of platelet function to healthy controls. In addition, patients and healthy controls were selected according to the following inclusion criteria: platelet count >100 × 10^9^ L^−1^, controls of fertile age, patients who are planning to have children, and patients without hormonal therapy prior to platelet function analysis. Pregnancy, postmenopausal women, infection or inflammation in the last 4 weeks before platelet function analysis, liver dysfunction, renal dysfunction, plasmatic coagulation protein disorders, excessive physical activity 24 h before platelet function analysis and medication within two weeks before platelet function analysis with acetylsalicylic acid (ASA), anticoagulation, NSAIDs, proton pump inhibitors, phosphodiesterase inhibitors, desmopressin, antibiotics, chemotherapeutics, plasma expanders and phytopharmaceuticals were exclusion criteria. With these selection criteria, none of the investigated controls and patients had preanalytical apparent disturbed platelet function at the time of blood draws.

Detailed patient medical history regarding miscarriage or infertility anamnesis and associated laboratory analysis is shown in Table 2. In order to retrospectively determine a possible pregnancy outcome after discharge and with regard to a successful live birth, we contacted the patients’ practitioners.

The local ethics committee approved our study in accordance to the ethical standards laid down in the Declaration of Helsinki (reference number EK056/15).

All included patients were differentiated according to their pregnancy loss history. A total of 79% (*n* = 164) of 208 patients evaluated had a positive miscarriage history. Among the 139 patients without SPS, 81% (*n* = 113) reported at least one miscarriage. Of the women suffering from SPS, 74% (*n* = 51) reported at least one miscarriage. Statistically significant differences in the miscarriage history cannot be found in patients with or without SPS (Table 2). A total of 38 patients had an internal and endocrine disorder (hypothyroidism, diabetes mellitus, or microprolactinome/hyperprolactinemia). Hypothyroidism was diagnosed as the most frequent disease (*n* = 33; with SPS *n* = 15, without SPS *n* = 18). The recorded endocrine and gynecological diagnoses include endometriosis, polycystic ovarian syndrome, oligomenorrhea, hyperandrogenemia and corpus luteum insufficiency (*n* = 29; with SPS *n* = 9, without SPS *n* = 20). The most frequent disorder was endometriosis (*n* = 19; with SPS *n* = 5, without SPS *n* = 14). Uterine and ovarian abnormalities were observed (*n* = 49; with SPS *n* = 17, without SPS *n* = 32). These include uterus myomatosus, uterus duplex, uterus septum, ovarian, endometrial cysts, tube occlusion, cervical stenosis and hydrosalpinx. Uterus myomatosus was found in 21 women (with SPS *n* = 7, without SPS *n* = 14). As genetic disorders are also causes of infertility or pregnancy loss, the methylenetetrahydrofolate reductase (MTHFR), Factor-V-Leiden or Prothrombin mutation was found in 39 patients (with SPS *n* = 10; without SPS *n* = 29). The most frequent disorder was MTHFR mutation, but no SPS was found (heterozygous *n* = 19; homozygous *n* = 6). Moreover 10 patients suffered from antiphospholipid syndrome and 5 patients were documented to have an immunovasculitis. For only 2 patients, the occurrence of a hyperlipidemia was reported. In sum, we could not find any statistical significance among these recorded diagnoses.

### 2.2. Measurement of Platelet Hyperaggregability

Platelet function was evaluated by testing aggregation responses to adenosine diphosphate (ADP) and epinephrine by optical light transmission aggregometry (Apact 4S Plus Aggregometer Analyser, Rolf Greiner BioChemica, Flacht, Germany), according to the method established by Born and Cross and modified by Mammen [18,27]. To minimize the preanalytical activation of platelets and clotting proteins, citrated (sodium citrate 0.11 mol L^−1^) whole blood samples were drawn by venopuncture through wide-bore needles without tourniquet. Platelet aggregation was performed within 4 h after blood sample collection. All participating patients and healthy controls had initial platelet counts within the reference range (Table 3). The aggregometry analysis was performed using native platelet-rich plasma (PRP; without platelet count adjustment, because there is no accepted lower limit of PRP platelet counts for aggregation testing) and platelet-poor plasma (PPP) [24,28,29,30]. Platelets in PRP are stimulated to aggregate by platelet aggregation inductors (e.g., ADP and epinephrine) and light transmission intensity is measured in relation to PPP transmission intensity. Prior to ADP and epinephrine stimulation, light transmission in PRP was defined as 0% (no platelet aggregation) and 100% in PPP (maximum platelet aggregation). The percentage light transmission in the stimulated PRP is the induced aggregability of the platelets in relation to the concentration of the inductor used [24]. The microscopic assessment of platelet morphology in citrated whole blood smear was carried out to exclude platelet plugs and spontaneous aggregation prior to aggregation analysis. However, our laboratory practices were in line with recognized guidelines regarding the laboratory investigation of heritable disorders of platelet function and published recommendations for the standardization of light transmission aggregometry [31,32]. Within the scope of the present work, the preparation of PRP and PPP were performed according to the recommendations of the Medical Standards Committee of the German Institute for Standardization (Deutsches Institut für Normung e.V. (DIN), Berlin, Germany) for the performance of a platelet function test [33]. According to this, we did not adjust the platelet count of PRP during the assessment of PRP quality. One additional reason is the fact that, in all samples, the platelet counts in PRP samples were not lower than 150 × 10^9^ L^−1^.

### 2.3. Identification of Patients with Platelet Hyperaggregability

At present, there are no validated reference intervals in the literature for platelet aggregation by multiple concentrations of agonists [24]. Therefore, we evaluated dose-dependent platelet aggregation patterns of ADP and epinephrine by stimulating native PRP samples in 29 age-matched healthy, non-pregnant, fertile women as a benchmarking diagnostic control group. Platelet aggregation was investigated in healthy controls in the same way as in patient analysis: PRP stimulation with ADP and epinephrine at four different concentrations according to the recommended concentrations for testing: ADP (µmol L^−1^): 10, 2, 1, 0.5; epinephrine (µmol L^−1^): 50, 10, 1, 0.5 [24]. Diagnostically, the dose-independent platelet aggregation indicates a sticky platelet syndrome: type 1: hyperaggregability with epinephrine and ADP; type 2: hyperaggregability only under epinephrine stimulation; type 3: hyperaggregability only under ADP stimulation [34]. The most common SPS type is the SPS type II, followed by the SPS type I (approximately 30%). SPS type III is rarely found and diagnosed in only about 1% of SPS patients [35,36]. The different types improve clinically identical and there are no differences in treatment. Platelet aggregation patterns in patients compared to healthy controls were determined as hyperaggregation under the following published conditions: (1) significantly increased median platelet aggregation at low-dose epinephrine (0.5 µmol L^−1^) when compared to median low-dose epinephrine aggregation in healthy controls (>26%); (2) significantly decreased Δ-ADP (delta-ADP) and Δ-epinephrine (delta-epinephrine) aggregation patterns (difference of platelet aggregation between the highest versus the lowest concentration of each agonist) in patients compared to healthy controls, median Δ-platelet aggregation response significantly lower than 68% for ADP and 57% for epinephrine. The ADP-induced aggregation was not considered as significant for potential platelet hyperaggregability due to the presence of dose-dependent ADP-induced platelet aggregation patterns in all the study cohorts (Table 3).

### 2.4. Statistical Analysis

Owing to the skewed distribution of the aggregation parameters, data are given as median and range, and shown graphically by box-and-whisker plots. Comparisons of platelet aggregation between two different groups were conducted with the Mann–Whitney U-test. All values, including outside values as well as far out values, were included. Receiver under the curve (ROC) analysis was carried out to determine the diagnostic sensitivity and specificity of aggregation responses for ∆-epinephrine aggregation (difference in high- and low-dose epinephrine-induced platelet aggregation). The prediction of platelet hyperaggregability by inductors of platelet function (ADP and epinephrine) was conducted with multiple linear regression analysis. The prediction of pregnancy outcome was investigated using multiple logistic regression analysis. *P*-values less than 0.05 were considered as statistically significant. All statistical analyses were performed using MedCalc statistical software version 11.4.2 (Mariakerke, Belgium).

## 3. Results

### 3.1. Dose-Dependent Platelet Aggregation in Healthy Controls

In healthy controls, normal platelet aggregation in native PRP after stimulation by highest ADP concentration (10 µmol L^−1^) was determined as 84% (median aggregation; range 50–120%) and 86% (median aggregation; range 24–120%) by highest epinephrine stimulation (50 µmol L^−1^), respectively. The median platelet aggregation responses to lowest ADP and epinephrine concentrations were significantly and dose-dependently decreased (0.5 µmol/L ADP 16%, range 10–47%, vs. ADP 10 µmol L^−1^, *p* < 0.0001; 0.5 µmol L^−1^ epinephrine 26%; range 8–55%, vs. 50 µmol L^−1^ epinephrine, *p* < 0.0001) (Figure 1a). The median ∆-platelet aggregation response defined by the difference between highest and lowest concentrations of ADP and epinephrine was 68 and 57%, respectively (Table 3). Decreased Δ-aggregation patterns (difference in platelet aggregation between the highest versus the lowest concentration of each agonist) were associated with absence of dose dependence in patients compared to healthy controls. Only ∆-epinephrine aggregation response clearly differentiated healthy controls and SPS patients, whereas paradox significant lower epinephrine (0.5 µM) aggregability was found in non-SPS patients (Table 3). These data clearly demonstrate that our healthy controls are not affected by platelet hyperaggregability. Platelet aggregation response in these healthy controls was used to determine platelet hyperaggregation in patients evaluated for potential IVF due to infertility evaluation or the diagnostic assessment of pregnancy loss history (Table 2 and Table 3).

### 3.2. Dose-Independent Epinephrine Induced Platelet Aggregation in Patients with Pregnancy Loss or Infertility as Compared with Healthy Controls

Platelet function was assessed in 208 consecutive patients between 2001 and 2012 as part of the routine clinical evaluation for IVF due to infertility or diagnostic assessment of pregnancy loss history. Of these patients, 139 patients showed a physiological dose-dependent pattern of platelet aggregation with decreasing concentrations of ADP and epinephrine (Table 3 and Figure 1b). In this subgroup, platelet hyperaggregability could be excluded (as described above in comparison with healthy controls). Patients with normal dose-dependent platelet function demonstrated a median ∆-epinephrine aggregation response (difference of the highest vs. lowest epinephrine stimulation) of 59% (range 7–88%), similar to the Δ-epinephrine aggregation of 57% (median, range 8–106%) in healthy patients.

On the contrary, 69 patients displayed a pathologic platelet aggregation pattern (Figure 1c). The median ∆-epinephrine aggregation response in these 69 patients with identified dose-independent platelet hyperaggregation was significantly lower compared to patients without SPS and healthy controls (7%; range −21 to 43% vs. 59%; range 7–88%, all *p* < 0.0001 and vs. 59%; range 7–88%, *p* < 0.0001) (Table 3 and Figure 1d). This pattern is typical for SPS type II. Platelet hyperaggregability induced by ADP and thus a possible SPS type I or type III was not observed in any of the SPS patients (Table 3). Moreover, multiple linear regression analysis to analyze the prediction of platelet hyperaggregability by ADP and epinephrine dose-dependent platelet aggregation analysis showed only ∆-epinephrine to be a significant indicator of platelet hyperaggregability (Table 4).

### 3.3. Prevalence of SPS in Patients with Pregnancy Loss or Infertility

Over the 11-year period, we found a high prevalence of SPS type II (33.2%) in our study cohort (Table 2). Interestingly, we observed a trend towards a higher number of pregnancy loss in most non-SPS patients. A total of 41% of non-SPS patients reported at least three miscarriages. In patients with SPS, 37% (*n* = 19) reported two miscarriages (differences were not significant).

### 3.4. Platelet Aggregability and ASA Therapy

Of 69 patients with SPS analyzed, 13 were not treated with low-dose ASA. However, reasons for this could not be found in all women. One patient was documented to have an ASA allergy. Three of these patients became pregnant, but only one could complete the pregnancy. In eight patients, it was not possible to determine whether pregnancy had occurred by contacting the patients’ practitioner. Thus, not every patient could retrospectively receive a definitive statement on pregnancy and/or live birth from the interviewed physicians. SPS patients with low-dose ASA therapy (a therapeutic inhibitor of platelet aggregation) (*n* = 56) showed a trend to improved pregnancy outcome compared to SPS patients without low-dose ASA (*n* = 13) (pregnant SPS patients *n* = 32, 57.1%, vs. pregnant non-SPS patients *n* = 3, 23.1%; live births *n* = 18, 56.3%, vs. *n* = 1, 33.3%) (Table 5). Thus, pregnant SPS patients with low-dose ASA revealed a tendency towards a reduced rate of pregnancy loss than pregnant SPS patients without low-dose ASA (*n* = 9, 28.3%, vs. *n* = 2, 66.7%). However, we could not find any statistically significant associations between ASA therapy and pregnancy outcome.

Moreover, because our data indicated that a higher number of identified pregnant SPS patients with low-dose ASA had revealed a tendency towards a reduced rate of pregnancy loss than pregnant patients without low-dose ASA, we conducted a multiple logistic regression analysis. In line with the description based on Table 5, the prediction of pregnancy outcome by prophylactic administration of ASA (100 mg per day) strongly depends on platelet function. The only predictor with a highly significant odds ratio (12.6806) is the laboratory diagnostic presence of platelet hyperaggregability, clinically known as SPS in patients in our study cohort. Further secondary diagnoses of the patients are not predictive, such as miscarriage history, endocrine dysfunction, obesity, smoking, genetic coagulation disorder and uterine/ovarian abnormalities (Table 6).

### 3.5. Sensitivity and Specificity of Δ-Epinephrine Platelet Aggregability in Patients Evaluated for Potential In Vitro Fertilization (IVF) Due to Infertility Evaluation or Diagnostic Assessment of Pregnancy Loss History

We performed a receiver operating characteristic (ROC) curve analysis, as depicted in Figure 2 (sensitivity vs. 100–specificity). The ROC analysis revealed a favorable cut-off value of ≤32% for Δ-epinephrine aggregation response distinguishing SPS patients from healthy controls and non-SPS patients with a sensitivity of 95.7% and a specificity of 95.2% (Figure 2 and Table 7). The AUC for Δ-epinephrine aggregation was 0.985 (95% CI 0.960–0.996) (Table 7). If the platelet aggregation analysis shows a value of less than 32% for the Δ-epinephrine aggregation response, it is highly probable that SPS can be diagnosed and initiating therapy with low-dose ASA may be considered.

## 4. Discussions

We herein report for the first time a surprisingly high prevalence of platelet hyperaggregability in patients with infertility and unexplained pregnancy loss history (33.2% for the 11-year period prevalence) when comparing to the study of Bick and Hoppensteadt [6]. To our knowledge, this high prevalence is only exceeded in patients with chronic kidney disease (CKD), such as chronic hemodialysis (82%) or renal transplantation (67%) [24]. Remarkably, no pathological platelet aggregation could be found in healthy, non-pregnant female individuals of fertile age. However, the definite prevalence of the SPS in the unselected total population has not yet been fully clarified [37]. Due to the limited number of patients studied, the exact prevalence and incidence of the SPS in the general population is not answered in detail by this study. It is considered that the possibility of SPS-diagnosis is less recognized in patients with unexplained or unprovoked thromboembolic diseases and in women with repeated miscarriages [6]. 

In our study, platelet hyperaggregability after both ADP and epinephrine (type 1 SPS) and platelet hyperaggregability after ADP only (type 3 SPS) were not found. According to Kubisz et al. [34], type 2 (platelet hyperaggregability after epinephrine only) is most common, followed by type 1, whereas type 3 is rare. However, it is important that this classification is based on laboratory criteria. In addition, no relation or differences to the clinical manifestation, treatment, or prognosis of patients were observed among the types so far. Interestingly, type 2 is the most frequent variant of SPS in white populations, whereas type 1 is the most frequent variant in Mexican mestizos [35]. However, despite the clear clinical definition and strong evidence of familiar occurrence, published results so far failed to identify a single genetic defect responsible for SPS. In sum, the laboratory heterogeneity of the syndrome with three clearly distinct types might suggest that SPS might have multifactorial genetics. Furthermore, because SPS diagnosis is made solely on the clinical and laboratory criteria and not on genetic testing, inherited and acquired changes of platelet aggregation cannot be ruled out in the currently diagnosed patients. It is therefore also likely that the SPS itself does not cause thromboembolic events, similar to protein C resistance or deficiency, but only predisposes to them. It is suspected that an additional factor is required for the development of a clinically relevant thromboembolic event [5]. 

Furthermore, SPS patients with low-dose ASA therapy showed a trend to improved pregnancy outcome compared to SPS patients without low-dose ASA. This finding is in line with the fact that SPS could be responsible for a large number of unexplained thromboembolic events of unknown causes in selected populations such as patients with pregnancy loss or diseases associated with increased vascular morbidity such as CKD [5,24]. Due to the low number of patients both without ASA therapy but successful pregnancy, the diagnostic and clinical relevance of less successful pregnancy in these patients might be premature at this stage of this explorative statistical analysis. However, further research should aim at investigating more SPS patients without ASA in regard to pregnancy outcome and to clarify the exact pathogenic and therapeutic role of platelet hyperaggregability in clinical setting. Bick described that SPS could be diagnosed in 21% of patients with miscarriages as well as coagulation disorders [5]. Only the antiphospholipid syndrome occurring with a frequency of 67% was more frequent [6]. Other reports also suggest similar frequencies. According to Kubisz et al., SPS has been identified in approximately 21% of unexplained arterial thrombotic episodes, regarded to be the most common thrombophilia in arterial thrombosis and 13.2% of unexplained venous thromboembolism [38]. Therefore, SPS is described as the most frequent hereditary prothrombotic platelet disorder in vascular morbidity [34]. 

The investigation of platelet function analysis in our retrospectively evaluated patients showed a pathologic platelet aggregation pattern in 69 patients. The median ∆-epinephrine aggregation response in these patients with identified dose-independent platelet hyperaggregation was significantly lower compared to patients without SPS and healthy controls. This pattern is typical for sticky platelet syndrome type II. Although some of these patients with SPS had MTHFR, Factor–V–Leiden or Prothrombin mutations, none had any recorded history of clinical thromboembolism other than the gynaecological disease. Although there was a high number of successful pregnancy outcomes in patients with SPS and even more in SPS plus low-dose ASA therapy compared to patients without SPS or ASA therapy, it did not reach statistical significance. It is likely that platelet hyperaggregability itself does not cause thromboembolic events, similar to activated Protein C resistance or Protein C deficiency, but only predisposes to them and a second hit may be required to develop overt clinical disease. Our finding is in accordance with the fact that Mammen and Bick suggested platelet hyperaggregability in the context of SPS as the cause of thromboembolic events, which could not be explained by any of the established risk factors (Protein C/S deficiency, Factor V Leiden mutation, antiphospholipid syndrome, etc.) [5,6]. In the course of time, this change in platelet function was associated with a variety of clinical pictures (myocardial infarction without congenital heart defect, recurrent apoplexy despite optimal oral anticoagulation, ischemic optic neuropathy, thromboembolic complications after kidney transplantation [1,8,9,10,11]). The thrombotic defects associated with fetal loss are thought to occur due to thrombosis of early placental vessels, with a peak in the first trimester, but small peaks also occur in the second and third trimesters [5,6]. It appears that the earlier the pregnancy, the smaller the placental and uterine vessels and, therefore, the greater the propensity to undergo partial or total occlusion by thrombus formation. Thrombotic occlusion of placental vessels, both venous and arterial, preclude adequate nutrition and, thus, viability of the fetus [5,6].

The therapy of SPS is based on the inhibition of platelet aggregation with low-dose ASA. However, standardized guidelines do not yet exist and the decision for treatment is considered individual [37]. In most patients, a low dosage of ASA of 80–100 mg per day is sufficient to normalize platelet hyperaggregation [39]. ASA is effective in both therapy and the prevention of thrombosis. When ASA is discontinued, however, pathological platelet function can be measured again. Alternatively, therapy with ADP inhibitors is also possible. In 2013, Velázquez–Sànchez–de–Cima et al. observed a very good therapeutic effect in SPS patients both with ASA and with a combination therapy with ASA and clopidogrel [40]. Other anticoagulants that act on plasmatic coagulation [oral anticoagulants] are ineffective and do not prevent the occurrence of thromboembolic events in SPS [35].

In addition, low-dose ASA treatment significantly improves ovarian responsiveness, uterine and ovarian blood flow velocity, and implantation and increases pregnancy rates and birth rates in IVF patients [41,42]. However, despite the better pregnancy outcome of SPS patients with ASA therapy, we could not find any statistically significant associations between ASA therapy and pregnancy outcome in our cohort. Similar conclusions were found in evaluations of ASA therapy in women undergoing IVF treatment. ASA did not support but showed a trend of improvement of clinical pregnancy outcome [43,44].

Mammen et al. had already suspected a hereditary cause of SPS by family testing at that time [15]. Recently, SPS is characterized as an autosomal dominant trait [37]. However, no distinct molecular genetic cause for SPS has been found to date [37]. Nevertheless, in several studies different genes were investigated, but no exact mutation could be detected which is responsible for the development of SPS. Kotuličová et al. investigated the glycoprotein 6 (GP6) polymorphism in SPS patients. It was found that there was a significant association between GP6 gene polymorphism and thromboembolic events in SPS patients [45]. Sokol et al. showed a significantly increased incidence of these GP6 polymorphisms in SPS patients with miscarriages [20]. Furthermore, Ruiz–Argüelles et al. analyzed the glycoprotein IIIa PI A1/A2 polymorphism in conjunction with the SPS, but no significant association was found [46]. Sokol et al. investigated SPS patients with miscarriages and controls for genetic polymorphism and Gas6/PEAR1 gene polymorphisms could be described [16]. Yee et al. showed in healthy individuals with platelet hyperaggregability an association with GNB3 polymorphisms [47].

Some methodical limitations regarding the platelet function analysis provided need to be taken into account. PRP does not contain all the platelets. Activated platelets or large platelets may be lost during the enrichment. The PRP analysis always takes place in an artificial environment. For example, erythrocytes and leukocytes are missing as endogenous sources for ADP and adenosine triphosphate (ATP) [48]. Furthermore, the definition of platelet hyperaggregability and the laboratory methods used are not standardized. Furthermore, platelets ex vivo and during phlebotomy could be heavily affected by preanalytical errors [12,30]. However, our laboratory practices were in line with recognized guidelines regarding the laboratory investigation of heritable disorders of platelet function and published recommendations for the standardization of light transmission aggregometry [31,32].

Light transmission aggregometry (LTA) measures the transmission of light through a sample of platelets in various suspensions such as platelet-rich plasma (PRP), washed platelets or gel-filtrated platelets. What complicates this methodology is that LTA results can be strongly influenced by the time between blood collection and analysis, platelet count, but also size, hematocrit, storage and measurement temperature, depending on the test system. Despite the complexity of platelet stabilization before testing, however, blood samples for LTA should be drawn into sodium citrate, buffered anticoagulant. In various standardization guidelines, venous citrated plasma and PRP preparation is recommended as an anticoagulant [49]. Nevertheless, this circumstance is contrary to the fact that stability of blood sample may be improved if the blood is anticoagulated without citrate, as citrate complexes calcium and platelets need calcium ions to function normally. Therefore, other anticoagulants can be similarly used and sometimes should be preferred, e.g., hirudin. Hirudinized blood contains the normal concentration of Ca^2+^ and Mg^2+^ [50]. However, citrate anticoagulated blood is still used in most test methods. Ultimately, the aggregation formation as it is registered in aggregometers is an artefact and only very indirectly corresponds to the complex in vivo process of platelet function. Thus, there is still no generally accepted ideal measure of platelet activation [50]. 

According to Cattaneo and coworkers, the platelet count of PRP samples should not be adjusted to a standardized value with autologous PPP [32]. Thus, recent studies demonstrated that platelet counts in PRP within the range that is observed in PRP samples from subjects with normal platelet count in whole blood do not affect the results of LTA studies [32]. In this line, abnormalities of platelet aggregation were more frequent using adjusted platelet count both in controls and patients [28]. Overall, it must therefore be stated that there is still controversy concerning whether the platelet count should be adjusted or not [51,52]. On the one hand, it has been argued that in vitro aggregation is basically influenced by the platelet count in PRP and thus platelet count adjustment is recommended. On the other hand, PPP may contain substances affecting platelet function that are released by platelets or other blood cells during centrifugation of blood samples, which is necessary to obtain PPP [50]. However, citrate plasma is not the optimal medium for platelet function testing. Therefore, further studies should be conducted to investigate the diagnostic value of other high-quality analytical strategies based on gel-filtered platelets or/and plasma switch in LTA analysis. Moreover, specialized scientific assessments such as enzymes of homocysteine metabolism, soluble P-selectin, E-selectin or pentraxin 3, plasminogen activator inhibitor (PAI)-1 4G/5G insertion–deletion mutations and coagulation factor XIII Val34Leu polymorphism or flow cytometry for detection of platelet-specific activation markers (such as P-selectin and fibrinogen binding or others like marker antibodies CD62, CD41, PAC1 (activated GP IIb/IIIa) and CD154) may potentially be explored. The analysis of these parameters might provide more objectivity and stability of results in an analysis of platelet function.

In addition, there are no studies to date that investigate the prevalence of SPS in the unselected total population; it is probable that the frequency of SPS is lower here than in the patient groups with thromboembolic events investigated to date [34]. For example, Kubisz et al. reported an SPS prevalence of 14% based on other patient cohorts with general thromboembolic events [25]. In contrast, Ruiz–Arguelles found SPS in up to 60% of primarily hypercoagulable patients [53,54].

## 5. Conclusions

In conclusion, our study shows for the first time that SPS is a frequent and potentially miscarriage-inducing disease that can be significantly prevented by simple low-dose ASA therapy. Decreased ∆-epinephrine-induced platelet aggregation response and hyperaggregable platelets indicate diagnostic relevance in gynecology and reproductive medicine to identify SPS. Low-dose ASA therapy in SPS patients potentially represents a promising strategy to improve pregnancy outcome. Based on a large observational cohort of consecutive patients, we provide clinical evidence that SPS may participate in the pathophysiology of unexplained pregnancy loss and infertility. However, the clinical relevance of SPS in infertility and diagnostic assessment of pregnancy loss history warrants further investigation.

## Figures and Tables

**Figure 1 jcm-08-01328-f001:**
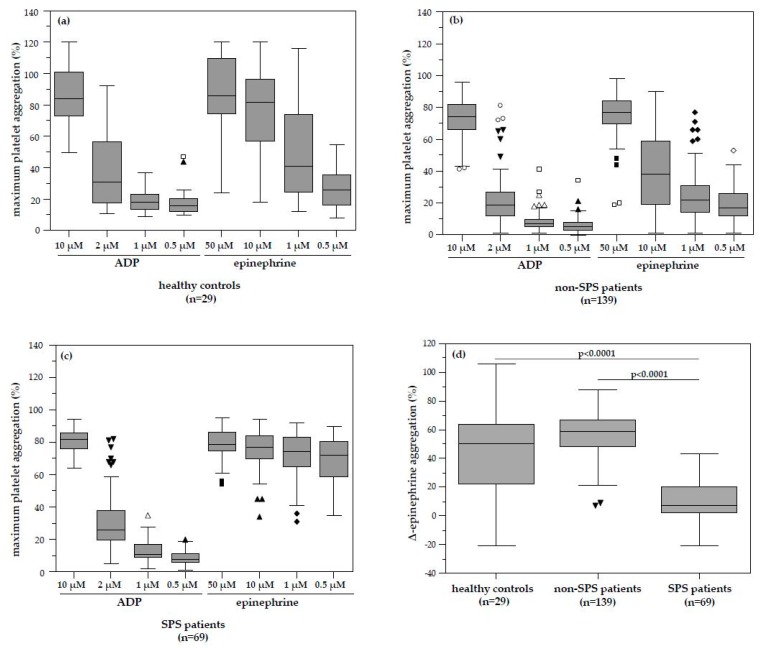
Platelet aggregability in healthy controls and patients. (**a**) Platelet aggregation in 29 healthy controls showed a physiological dose-dependent decrease with decreasing concentration of the added inductors ADP and epinephrine. (**b**) Physiological dose-dependent pattern of platelet aggregation in healthy controls was also found in 139 of the 208 patients. (**c**) Pathological platelet aggregation pattern was found in 69 patients. The platelet aggregation was significantly increased independent of the lowest added epinephrine concentration (0.5 µmol L^−1^) compared to non-SPS patients. (**d**) The median Δ-epinephrine aggregation was found to be 57% in healthy controls. Patients without platelet hyperaggregability revealed equivalent Δ-epinephrine aggregation compared to healthy patients (59% vs. 57%; *p* = 0.8190). ADP, Adenosine diphosphate; SPS, Sticky platelet syndrome.

**Figure 2 jcm-08-01328-f002:**
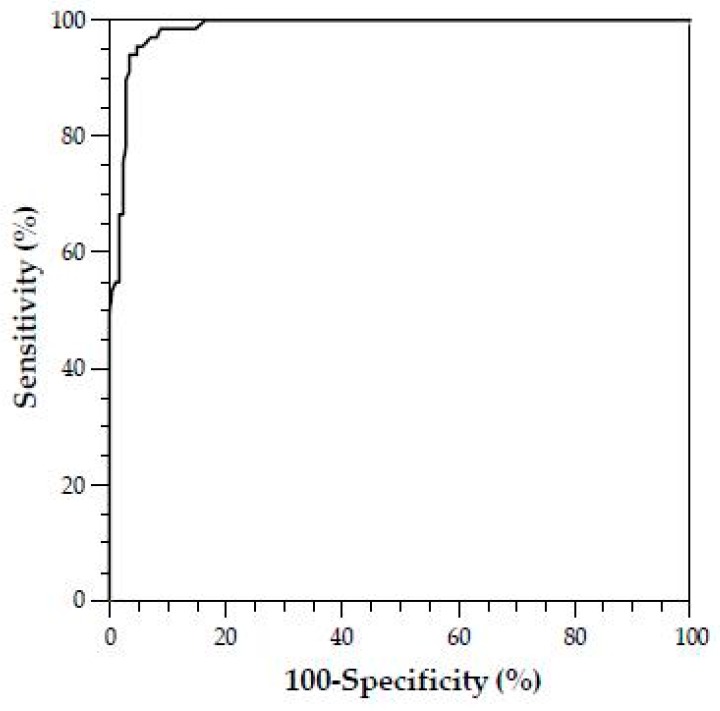
Diagnostic value of Δ-epinephrine aggregation in the assessment of SPS. Receiver operating characteristic curve (ROC) analysis showing a favorable cut-off value of ≤32% for Δ-epinephrine aggregation in distinguishing SPS patients from healthy controls. The area under the curve (AUC) was calculated with a 95% confidence interval (CI).

**Table 1 jcm-08-01328-t001:** Age of healthy controls and patients.

Parameter	Healthy Controls	Patients	* *p*
Number *n*	29	208	
Age (years)	36 (24–49)	34 (20–45)	n.s.

Median and range (in parenthesis) are given. ***** Significance between healthy controls and patients was assessed using the Mann–Whitney U-test. n.s., not significant.

**Table 2 jcm-08-01328-t002:** Baseline patient characteristics and clinical medical history.

Parameter	All Patients	SPS	Non-SPS	* *p*
Number *n* (%)	208	69 (33.2%)	139 (66.8%)	n.s.
Pregnancy loss history *n* (%)	164 (79%)	51 (74%)	113 (81%)	n.s.
No pregnancy loss history *n* (%)	44 (21%)	18 (26%)	26 (19%)	n.s.
** Obesity *n* (%)	40 (19%)	8 (12%)	32 (23%)	n.s.
Nicotine consumption *n* (%)	31 (15%)	7 (10%)	24 (17%)	n.s.
Internal and endocrine disorder n (%)	38 (18%)	15 (22%)	23 (17%)	n.s.
Gynecological and endocrine disorder *n* (%)	29 (14%)	9 (13%)	20 (14%)	n.s.
Genetic disorder *n* (%)	39 (19%)	10 (14%)	29 (21%)	n.s.
Uterine/ovarian disorder *n* (%)	49 (24%)	17 (25%)	32 (23%)	n.s.
Antiphospholipid syndrome *n* (%)	10 (5%)	3 (7%)	7 (5%)	n.s.
Immunovasculitis *n* (%)	5 (2%)	3 (7%)	2 (1%)	n.s.
Age (years)	34 (20–45)	34 (20–45)	34 (23–43)	n.s.
TSH (mU/mL)	1.6 (0.01-6.3)	1.7 (0.18-6.3)	1.5 (0.01–6.1)	n.s.
Protein C activity (%)	111 (56–208)	102 (56–150)	116 (72–208)	n.s.
Protein S activity (%)	86 (33–140)	89 (33–140)	86 (53–140)	n.s.
Activated protein C resistance (%)	2.5 (1.3–3.0)	2.6 (1.3–3.0)	2.4 (1.3–3.0)	n.s.
Rubella hemagglutination (titre)	1:64 (1:8–1:265)	1:64 (1:16–1:256)	1:32 (1:8–1:265)	n.s.

For quantitative variables, median and range (in parenthesis) are given. Percentages in parenthesis refer to the total number of patients in the respective groups. ***** Significance between SPS and non-SPS patients was assessed using the Mann–Whitney U-test (for quantitative variables) or the chi-square test (for categorical variables). ****** BMI value over 30 kg m^−2^. BMI, body mass index; TSH, Thyroid stimulating hormone; SPS, Sticky platelet syndrome, n.s., not significant.

**Table 3 jcm-08-01328-t003:** Platelet aggregation data in response to different ADP and epinephrine concentrations.

Platelet Aggregation	Healthy Controls (*n* = 29)	Patients			
		SPS (*n* = 69)	* *p*	Non-SPS (*n* = 139)	* *p*
Thrombocyte count [×10^9^ L^−1^]	163 (106–255)	243 (139–429)	n.s.	225 (138–425)	n.s.
ADP 10 µM [%]	84 (50–120)	82 (64–94)	n.s.	74 (41–96)	n.s.
ADP 2 µM [%]	31 (11–92)	26 (5–82)	n.s.	19 (1–81)	n.s.
ADP 1 µM [%]	18 (9–37)	11 (2–35)	n.s.	7 (1–42)	n.s.
ADP 0.5 µM [%]	16 (10–47)	8 (1–20)	n.s.	5 (0–34)	n.s.
∆-ADP (10–0.5 µM) [%]	68 (8–106)	71 (−15–53)	n.s.	70 (9–91)	n.s.
Epinephrine 50 µM [%]	86 (24–120)	79 (54–95)	n.s.	77 (19–98)	n.s.
Epinephrine 10 µM [%]	82 (18–120)	77 (34–94)	n.s.	38 (1–90)	n.s.
Epinephrine 1 µM [%]	41 (12–116)	74 (31–92)	n.s.	22 (1–77)	n.s.
Epinephrine 0.5 µM [%]	26 (8–55)	72 (35–90)	<0.0001	17 (1–53)	0.0004
∆-Epinephrine (50–0.5 µM) [%]	57 (8–106)	7 (−21–43)	<0.0001	59 (7–88)	n.s.

Platelet aggregation data in response to ADP and epinephrine concentrations as well as Δ-ADP/Δ-epinephrine aggregation response formed the basis for the definition of platelet hyperaggregability in patients. Median and range (in parenthesis) are given. ***** Significance between SPS and non-SPS patients vs. healthy controls was assessed using the Mann–Whitney U-test. ADP, Adenosine diphosphate; SPS, Sticky platelet syndrome, n.s., not significant.

**Table 4 jcm-08-01328-t004:** Prediction of platelet hyperaggregability.

Predictor	Coefficient	Standard Error	*p*
∆-epinephrine	−0.004071	0.001282	0.0017

The prediction of platelet hyperaggregability by inductors of platelet function was investigated using multiple linear regression analysis (R^2^ = 0.8179).

**Table 5 jcm-08-01328-t005:** Pregnancy outcome in patients with and without low-dose acetylsalicylic acid therapy.

		*n*	Pregnancy		Live Birth	
			Yes	No	Yes	No
**with low-dose ASA**	Patients	62	37 (60%)	25 (40%)	20 (54%)	11 (30%)
	Non-SPS	6 (10%)	5 (83%)	1 (17%)	2 (40%)	2 (40%)
	SPS	56 (90%)	32 (57%)	24 (43%)	18 (56%)	9 (28%)
**without low-dose ASA**	Patients	146	53 (36%)	29 (20%)	40 (76%)	11 (21%)
	Non-SPS	133 (91%)	50 (38%)	27 (20%)	39 (78%)	9 (18%)
	SPS	13 (9%)	3 (23%)	2 (15%)	1 (33%)	2 (67%)

ASA, acetylsalicylic acid; SPS, Sticky platelet syndrome.

**Table 6 jcm-08-01328-t006:** Prediction of pregnancy outcome.

Predictor	Odds Ratio	95% Confidence Interval	Coefficient	Standard Error	*p*
SPS	12.6806	3.6231–44.3807	2.54007	0.63915	0.0001

The prediction of pregnancy outcome was investigated using multiple logistic regression analysis.

**Table 7 jcm-08-01328-t007:** Diagnostic value of Δ-epinephrine aggregation in the assessment of SPS.

**AUC**	0.985
**95%CI**	0.960–0.996
**Sensitivity [%]**	95.7
**Specificity [%]**	95.2
**Optimal cut-off Δ-epinephrine [%]**	≤32

The Δ-epinephrine aggregation cut-off value of ≤32% reveals a high diagnostic sensitivity and specificity to distinguish SPS patients from healthy controls. AUC, area under the curve; CI, confidence interval.

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
