# Peer review of "High Prevalence of Sticky Platelet Syndrome in Patients with Infertility and Pregnancy Loss"

_jcm, 2019, doi:10.3390/jcm8091328_

Round 1

Reviewer 1 Report

Review comments:

After carefully reviewing this revised manuscript, most questions were addressed properly. The quality of the manuscript has been improved. There are some minor points that needs revision.

Minor:

Line 304-305. The subheading “Patients Evaluated for Potential IVF (IVF) " should be corrected to " Patients Evaluated for Potential in-vitro fertilization (IVF). The labels (a)-(d) in Figure 1 are suggested to put on the top-left of each data. The legends and footnotes should be revised which were not necessary to describe

some details of data, correlation, and author's comments. Those descriptions should mention in Results.

In Figure 1, it's not clear about the indication of those marks (triangle, square and circle ) on the figure ?

Author Response

Response to Reviewer 1 Comments

Round 2

Point-by-point response to the reviewer:

Thank you very much for the thorough and fair review of our manuscript.

 Reviewer:

After carefully reviewing this revised manuscript, most questions were addressed properly. The quality of the manuscript has been improved. There are some minor points that needs revision.

Response:

We thank the reviewer for the previous, very helpful comments and hope that our revised manuscript is now acceptable for publication.

Line 304-305. The subheading “Patients Evaluated for Potential IVF (IVF) " should be corrected to " Patients Evaluated for Potential in-vitro fertilization (IVF).

Response:

We thank the reviewer for this remark and his/her very careful attention. We have corrected this subheading as suggested by the reviewer (see page 9; lines 317-318).

The labels (a)-(d) in Figure 1 are suggested to put on the top-left of each data.

Response:

We thank the reviewer for this remark and his/her very careful attention (see page 7; Figure 1).

The legends and footnotes should be revised which were not necessary to describe some details of data, correlation, and author's comments. Those descriptions should mention in Results.

Response:

We fully agree with the referee concern that legends and footnotes should be revised. We mentioned some of these legend notes directly in the results (see Tables 3 and 5 as well as Figure 1).

In Figure 1, it's not clear about the indication of those marks (triangle, square and circle ) on the figure ?

Response:

We sincerely thank the referee for this remark. As described in the experimental section the marks in Figure 1 and each Box-and-Whiskers indicate outside values as well as far out values (see pages 5-6; lines 218-221).

Reviewer 2 Report

In the present retrospective study, the authors noticed high prevalence of platelet hyperaggregability pattern typical of SPS type II syndrome in women with pregnancy loss.

In particular, they demonstrated a significant reduction of delta-epinephrine‐induced platelet aggregation in SPS patients compared to patients without SPS and healthy controls. Moreover, they observed that low‐dose acetylsalicylic acid (ASA) therapy showed a trend to improve pregnancy outcome in SPS patients compared to SPS patients without ASA. Based on these findings, they conclude that SPS may participate in the pathophysiology of unexplained pregnancy loss and that low dose ASA therapy may be a useful strategy to prevent miscarriages in SPS patients.

Although the manuscript is interesting, there are major concerns that should be addressed. Listed below some specific comments.

It is well known that acquired conditions such as antiphospholipid syndrome or increased plasma levels of clotting factor VIII are associated with recurrent pregnancy loss. Have the authors evaluated additional markers for the thrombophilic state?

Hyperhomocysteinemia has been described as a risk factor for unexplained recurrent pregnancy loss. Moreover, it has also been associated with an increase in platelet activation. Although the most common enzyme defect associated with raised total homocysteine is due to a mutation in the gene for MTHFR (considered in this study), there are other causes of severe hyperhomocysteinemia among them homozygous deficiency of cystathionine-β-synthase, deficiency or impaired activity of methionine synthase. Have the authors considered homocysteine levels?

A complete panel of cholesterol profile should be reported, in order to make more understandable the data.

Plasminogen activator inhibitor (PAI)-1 4G/5G insertion-deletion mutations and coagulation Factor XIII Val34Leu polymorphism have been associated with impaired fibrinolysis and early pregnancy loss. The authors should at least determine these polymorphisms into the patient`s population.

Measurement of soluble P-selectin, E-selectin and PTX3 could highly improve the novelty of the manuscript since they seem to be related to platelets function and aggregation but data on the impact of ASA treatment are completely loss.

Author Response

Response to Reviewer 2 Comments

Round 1

Point-by-point response to the reviewer:

Thank you very much for the thorough and fair review of our manuscript.

 Reviewer:

In the present retrospective study, the authors noticed high prevalence of platelet hyperaggregability pattern typical of SPS type II syndrome in women with pregnancy loss.

In particular, they demonstrated a significant reduction of delta-epinephrineinduced platelet aggregation in SPS patients compared to patients without SPS and healthy controls. Moreover, they observed that lowdose acetylsalicylic acid (ASA) therapy showed a trend to improve pregnancy outcome in SPS patients compared to SPS patients without ASA. Based on these findings, they conclude that SPS may participate in the pathophysiology of unexplained pregnancy loss and that low dose ASA therapy may be a useful strategy to prevent miscarriages in SPS patients.

Response:

We thank the reviewer for his/her comments, which were very helpful and further helped to improve our manuscript.

Although the manuscript is interesting, there are major concerns that should be addressed. Listed below some specific comments.

It is well known that acquired conditions such as antiphospholipid syndrome or increased plasma levels of clotting factor VIII are associated with recurrent pregnancy loss. Have the authors evaluated additional markers for the thrombophilic state?

Response:

We appreciate this comment and have provided additional information in the experimental section and in Table 2 of the revised manuscript (see page 4; lines 164-166). However, the study included 208 female patients, who were admitted to the Department for Gynecological Endocrinology and Reproductive Medicine at the RWTH Aachen University Hospital between February 2001 and January 2012. All patients were presented for potential in-vitro fertilization (IVF) due to infertility evaluation (duration of infertility >1 year) or diagnostic assessment of unexplained pregnancy loss history.

In this clinical setting of patients for potential IVF therapy, the following haemostaseological analyses were performed in addition to the detailed clinical diagnostics as shown in Table 2 "Study design and patient characteristics": Protein C and protein S activities, activated protein C resistance as well as Factor-V-Leiden-, Prothrombin- and MTHFR- mutations. Nevertheless, we have analysed the data sets of the 208 patients for further thrombophilia-relevant coagulation parameters.

We were able to identify 10 patients with antiphospholipid syndrome (APS). We have listed these patients in the following table.

Parameter

All Patients

SPS

Non-SPS

APS

10

3

7

Lupus anticoagulant

3

1

2

Cardiolipin antibodies

4

0

4

Beta-2 glycoprotein-1- antibodies

2

1

1

Cardiolipin + beta-2 glycoprotein-1- antibodies

1

1

0

We listed the number of positive results for lupus anticoagulant as well as for the cardiolipin- and beta-2 glycoprotein-1- antibodies. Only 3 of these 10 APS patients had SPS. 7 of the 10 patients had antiphospholipid antibodies, while only 3 had a lupus anticoagulant. There was no statistical significance between the occurrence or absence of SPS. Additional coagulation analyses could not be subsequently found. Further coagulation tests could no longer be retested due to the clinical procedures performed up to 7-18 years ago. We assume that most patients were pre-diagnosed externally prior to admission to university IVF therapy planning. However, these data were not available.

Thus, due to the data available from the Department for Gynecological Endocrinology, no corresponding information was available for further hemostatic laboratory biomarkers dealing with factor XII deficiency, antithrombin deficiency, fibrinolytic defects or dysfibrinogenemias. However, further analyses or assessments of tissue plasminogen deficiency, tissue plasminogen activator (t-PA) deficiency, elevated plasminogen activator inhibitor type I (PAI-1) or even PAI-1 polymorphisms are not offered in clinical routine in Germany, so that these specialized assays were presumably not the subject of patients’ clinical procedure between 2001 and 2012.

In this regard, we would also like to emphasize that a re-evaluation of the multiple linear regression analysis as shown in Table 4 was not possible due to the limited number of cases for lupus anticoagulant and APS antibodies.

Hyperhomocysteinemia has been described as a risk factor for unexplained recurrent pregnancy loss. Moreover, it has also been associated with an increase in platelet activation. Although the most common enzyme defect associated with raised total homocysteine is due to a mutation in the gene for MTHFR (considered in this study), there are other causes of severe hyperhomocysteinemia among them homozygous deficiency of cystathionine-β-synthase, deficiency or impaired activity of methionine synthase. Have the authors considered homocysteine levels?

Response:

We thank the reviewer for mentioning this important aspect. Unfortunately, no homocysteine analysis was documented in the context of the retrospective analysis of patient data from a period from 2001 to 2012 at the University Hospital Aachen, which we could have presented. The aim of our evaluation was to identify SPS as an additional haemostaseological and prothrombogenic defect. We therefore would like to emphasize the retrospective exploratory nature of our work. However, the purpose of this statistical explorative analysis is to descriptively discuss the often in clinical routine unknown clinical relevance of sticky platelet syndrome. Understanding these aspects will help to better utilize the evidence to improve clinical decision-making. We assume that most patients were pre-diagnosed externally prior to admission to the University hospital. Unfortunately, we do not have access to external data, especially if the analyses were performed between 7 and 18 years ago.

Although we could not find data on homocysteine levels in the database, we would like to underline that we found ANCA diagnostics to help clarify a possible immunovasculitis. Of 208 patients, 5 were ANCA positive and 3 had SPS. These additional data on immunovasculitis have been added to Table 2 (see page 4; lines 164-166).

A complete panel of cholesterol profile should be reported, in order to make more understandable the data.

Response:

We sincerely thank the referee for this suggestion. We fully agree that platelet-lipoprotein binding has been claimed to be related to platelet reactivity. An assessment of lipoprotein metabolism was not routinely performed in the clinical treatment of patients in the Department for Gynecological Endocrinology. However, we were able to find clinically documented hyperlipidemia in only 2 out of 208 patients in the reassessment of secondary diagnoses. However, patients with overweight and adiposity-induced metabolic syndrome have been characterized by positive correlation to platelet activity markers (Sinzinger H. et al. Am J Hematol. 1991). Therefore, because our data indicate that a higher number of patients revealed obesity (n=40) and nicotine consumption (n=31) as shown in Table 2 we conducted and integrated the multiple logistic regression analysis into the manuscript (see result section subheading 3.4, Table 6). In line with the description above, the result show that prediction of pregnancy outcome by prophylactic administration of ASA (100 mg per day) depends on platelet function. The only predictor with a highly significant odds ratio (12.6806) found is the laboratory diagnostic presence of platelet hyperaggregability clinically known as SPS in patients with history of thromboembolism. Further secondary diagnoses of the patients are not predictive, such as miscarriage history, endocrine dysfunction, obesity, smoking, genetic coagulation disorder and uterine/ovarian abnormalities.

Predictor

Odds ratio

95%-Confidence interval

coefficient

Standard error

p-value

SPS

12.6806

3.6231-44.3807

2.54007

0.63915

0.0001

We have now described this fact in more detail in the results section and expanded the content of point 3.4 by the multivariate logistic regression analysis (see page 8-9; lines 306-316.

Plasminogen activator inhibitor (PAI)-1 4G/5G insertion-deletion mutations and coagulation Factor XIII Val34Leu polymorphism have been associated with impaired fibrinolysis and early pregnancy loss. The authors should at least determine these polymorphisms into the patient`s population.

Response:

We thank the referee for his/her thoughtful comments. As already mentioned, these specialized analyses were not performed at the time of data collection. We would like to note again at this point that these complex analyses are rarely performed together with a platelet function test in daily routine, in particular because both the mutation and polymorphism analyses are reserved for highly specialized laboratories and the platelet function tests for experienced laboratories. Fortunately, in cooperation between clinical reproductive medicine and laboratory medicine, we were able to obtain valuable data on platelet function over a very long period of time, which is likely to be very rare in this form, and from this perspective the data sets can still be regarded as clinically informative in the sense of explorative evaluation.

Measurement of soluble P-selectin, E-selectin and PTX3 could highly improve the novelty of the manuscript since they seem to be related to platelets function and aggregation but data on the impact of ASA treatment are completely loss.

Response:

We would like to thank the reviewer for this correct comment. Unfortunately, it is impossible to test for these markers afterwards. However, the reviewer is right with this very important point and all patients who receive aspirin therapy should be pre-diagnosed on these markers. We would like to point out that even in the antithrombocytic therapy of cardiovascular diseases, but also generally thromboembolic patients, a pre-test of the aspirin efficacy unfortunately is not done regularly, and Born aggregometry, the PFA-100 or the whole blood aggregometry (MULTIPLATE) analysis is frequently used.

Reviewer 3 Report

In this manuscript, Yagmur et al reported a high prevalence of platelet hyperaggregation in patients with pregnancy loss, and proposed Δ‐epinephrine aggregation as a potential diagnostic tool in these patients. This is a very interesting retrospective study, providing another underlying mechanism for unexplained pregnancy loss. But the following points need to be addressed before this work can be published in Journal of Clinical Medicine.

Major comments:

1). It is an interesting observation that low-dose ASA tended to improve the pregnancy outcome in SPS patients, but in the current version of manuscript, there is only 1 successful pregnancy in the control group (i.e. SPS patients without low-dose ASA). It is hard to draw any meaning conclusion from one patient, and thus more patients should be included in this retrospective study.

2). The authors only used PRP to evaluate the platelet aggregation, but both platelet-intrinsic factors and plasma can contribute to the platelet aggregation. Thus, gel-purified platelets should be tested (at least for some SPS patients and healthy control), to see the phenotype of dose-independent platelet aggregation in response to epinephrine stimulation is due to platelet-intrinsic effects or due to altered plasma. Alternatively, the authors can switch the plasma between SPS patients and healthy control (e.g. reconstitute gel-purified platelets from SPS patients with PPP from healthy control), and then test platelet aggregation.

3). In addition to platelet aggregation, it will enhance the quality of this manuscript, by performing some flow cytometry assays (e.g. fibrinogen binding to platelets and the abundance of active-form beta3 integrin) with some samples from SPS patients and healthy control, to test whether platelet hyperactivation is also prevalent in SPS patients.

4). For the platelet aggregation assays, the authors did not calibrate the concentrations of platelets in PRP among patients. It can be a potential drawback, as the platelet concentrations in SPS patients are almost 30% higher than healthy control. Typically, the same concentration of platelets in PRP should be used when comparing between samples. However, the PRP of SPS patients still exhibit a dose-dependent aggregation in response to ADP, thus acting as a “negative control” for their epinephrine phenotype. The authors should include this kind of discussion in the manuscript.

5). The authors should discuss why these SPS patients only exhibited the hypersensitivity to epinephrine, not ADP, in their platelet aggregation assays. What is the potential genetic basis behind this?

Minor comments:

1) The authors should cite some papers from animal studies (e.g.  Li et al, JCI, 2011, PMID: 22019589), which directly demonstrated the causal relationship between platelet hyperaggregability and pregnancy loss.

2) The authors should do a better proof-reading, to avoid the typos in the paper (e.g. life births in lines 35 and 292)

3) The subtitle of “Dose-independent platelet aggregation” (line 244) is inappropriate, as the SPS patients still exhibit dose-dependent aggregation in response to ADP.

Author Response

Response to Reviewer 3 Comments

Round 1

Point-by-point response to the reviewer:

Thank you very much for the thorough and fair review of our manuscript.

 Reviewer:

In this manuscript, Yagmur et al reported a high prevalence of platelet hyperaggregation in patients with pregnancy loss, and proposed Δepinephrine aggregation as a potential diagnostic tool in these patients. This is a very interesting retrospective study, providing another underlying mechanism for unexplained pregnancy loss. But the following points need to be addressed before this work can be published in Journal of Clinical Medicine.

Response:

We greatly appreciate the positive evaluation of our data and the suggestions, which helped to improve our manuscript.

Major comments:

1). It is an interesting observation that low-dose ASA tended to improve the pregnancy outcome in SPS patients, but in the current version of manuscript, there is only 1 successful pregnancy in the control group (i.e. SPS patients without low-dose ASA). It is hard to draw any meaning conclusion from one patient, and thus more patients should be included in this retrospective study.

 Response:

We would like to thank the reviewers for raising this important point. We agree that the diagnostic and clinical relevance might be premature at this stage of this explorative statistical analysis. We therefore would like to emphasize the exploratory nature of our work.

In the follow-up interviews of the patients' treating physicians, we could not find out why 13 SPS patients did not receive ASA, as indicated in the result section (see page 8, lines 291-296). Only one person was reported to have an aspirin allergy. However, of these 13 patients, 3 (23%) became pregnant, whereas only one successfully completed the pregnancy. In contrast, however, significantly more patients with SPS and ASA therapy were pregnant (n=32) (57%). 56% of these patients (n=18) had a successful pregnancy. Thus, in Table 5 it is described that relatively more SPS patients tended to become pregnant and more patients experienced a successful pregnancy than patients with SPS and without ASA therapy.

Moreover, because our data indicate that a higher number of identified pregnant SPS patients with low-dose ASA revealed a tendency towards reduced rate of pregnancy loss than pregnant patients without low-dose ASA we conducted and integrated the multiple logistic regression analysis into the manuscript (see result section subheading 3.4; pages 8-9; Table 6; lines 306-316).

In line with the description above, the result show that prediction of pregnancy outcome by prophylactic administration of ASA (100 mg per day) depends on platelet function. The only predictor with a highly significant odds ratio (12.6806) is the laboratory diagnostic presence of platelet hyperaggregability clinically known as SPS in patients with history of thromboembolism. Further secondary diagnoses of the patients are not predictive, such as miscarriage history, endocrine dysfunction, obesity, smoking, genetic coagulation disorder and uterine/ovarian abnormalities (multiple logistic regression analysis, Table 6).

Predictor

Odds ratio

95%-Confidence interval

coefficient

Standard error

p-value

SPS

12.6806

3.6231-44.3807

2.54007

0.63915

0.0001

We have now described this fact in more detail in the results section and expanded the content of point 3.4 by the logistic regression analysis.

It is worth noting that this observation is potentially consistent with the literature data on the efficacy of ASA therapy in platelet hyperaggregability

The therapy of SPS is based on the inhibition of platelet aggregation with acetylsalicylic acid (ASA). However, standardized guidelines do not yet exist and treatment is individual (Moncada B. et al. Hematology. 2013). In most patients, a low oral dosage of ASA of 80-100mg per day is sufficient to normalize laboratory and clinical hyperaggregation of platelets (Frenkel E.P. and Mammen E.F. Hematol Oncol Clin North Am. 2003). ASA is effective in both therapy and prevention of thrombosis. If this dosage is not sufficient, it can be increased up to 325mg per day. After discontinuation of ASA, pathological platelet function can be measured again. Alternatively, therapy with ADP inhibitors is also possible. In 2013, Velázquez-Sánchez-de-Cima et al. observed a very beneficial therapeutic effect in SPS patients both with ASA and with a combination therapy consisting of ASA and clopidogrel (Velázquez-Sánchez-de-Cima S. et al. Clin Appl Thromb Hemost. 2013). Other anticoagulants directly acting on plasmatic coagulation (DOAKs, direct oral anticoagulants) are ineffective and do not prevent the occurrence of thromboembolic events in SPS (Kubisz P. et al. Semin Thromb Hemost. 2013).

In summary, it is important to underline that our intention is to describe exploratively the potential interdependence of thrombocyte dysfunction in women with infertility and pregnancy loss in regard to prevalence and pregnancy outcome and to discuss the diagnostic and clinical relevance of platelet hyperaggregability and sticky platelets syndrome (SPS), respectively (see Introduction section, page 3, lines 103-107). We hope that our findings contribute to the understanding of platelet hyperaggregability as a useful tool for improved and deeper understanding the complex network of infertility and pregnancy loss.

In previous investigations of the causal associations of platelet hyperaggregability, miscarriages and SPS have frequently been reported together, but no investigation has so far investigated more closely at the pregnancy outcome in patients with SPS in line with prevalence analyses.

We have conducted an explorative and descriptive empirical study which shows for the first time that SPS is a frequent and potentially miscarriage-inducing disease that might be significantly prevented by simple low-dose ASA therapy.

However, further studies on this issue are urgently needed to clarify its exact pathogenic and therapeutic role in this setting.  

2). The authors only used PRP to evaluate the platelet aggregation, but both platelet-intrinsic factors and plasma can contribute to the platelet aggregation. Thus, gel-purified platelets should be tested (at least for some SPS patients and healthy control), to see the phenotype of dose-independent platelet aggregation in response to epinephrine stimulation is due to platelet-intrinsic effects or due to altered plasma. Alternatively, the authors can switch the plasma between SPS patients and healthy control (e.g. reconstitute gel-purified platelets from SPS patients with PPP from healthy control), and then test platelet aggregation.

 Response:

We appreciate this important comment and fully agree with the referee that among others both platelet-intrinsic factors and plasma can contribute to the platelet aggregation. We have provided additional information and discussed this issue in the discussion section (see page 11-12, lines 437-451).

We would like to emphasize the retrospective exploratory nature of our work. Due to the retrospective evaluation of patient data over a long period of time and from the routine clinical work up, respectively, an experimental approach with blood samples from patients and healthy controls is not applicable.

Particularly when testing the platelet function by LTA, it is well known that alterations in the sample material caused by errors in preanalytics can falsify the analytical results to such an extent that they are diagnostically unusable. However, the LTA technique is not standardized for testing of platelet aggregability, despite the fact that guidelines have been published. Therefore, LTA measures the transmission of light through a sample of platelets in various suspensions such as platelet-rich plasma (PRP), washed platelets or gel-filtrated platelets. What complicates this methodology is that LTA results can also be strongly influenced by the time between blood collection and analysis, platelet count, but also size, haematocrit, storage and measurement temperature, depending on the test system. Moreover, an international consensus panel recommends a short rest period and avoidance of smoking and caffeine prior to blood collection to mitigate the effects of endogen epinephrine release (Lit 6). Despite the complexity of platelet stabilization before testing, however, blood samples for LTA should be drawn into sodium citrate, buffered anticoagulant. In various standardization guidelines, venous citrated plasma and PRP preparation is recommended as an anticoagulant (Cattaneo M. et al. J Thromb Haemost. 2009). Nevertheless, this circumstance is contrary to the fact that stability of blood sample may be improved if the blood is anticoagulated without citrate, as citrate complexes calcium and platelets need calcium ions to function normally. Therefore, other anticoagulants can also be used and sometimes should be preferred, e.g. hirudin. Hirudinized blood contains the normal concentration of Ca2+ and Mg2+. However, citrate anticoagulated blood is still used in most test methods. Ultimately, the aggregeation formation as it is registered in aggregometers is an artefact and only very indirectly corresponds to the complex in vivo process of platelet function. Thus, there is still no generally accepted ideal measure of platelet activation that would indicate a state of „high risk“.

3). In addition to platelet aggregation, it will enhance the quality of this manuscript, by performing some flow cytometry assays (e.g. fibrinogen binding to platelets and the abundance of active-form beta3 integrin) with some samples from SPS patients and healthy control, to test whether platelet hyperactivation is also prevalent in SPS patients.

Response:

We fully agree with the referee that flow cytometry for detection of platelet-specific activation markers, such as markers for platelet activation (P-Selectin and fibrinogen binding) provides objectivity and stability of results.

As already mentioned in point 2, due to the nature of our study, which retrospectively evaluated patient data over a long period of time, an experimental approach with blood samples from patients and healthy subjects from the study is not possible.

We have had already considered exactly this procedure by using the following antibodies CD62P-FITC, CD41-PE, PAC1-FITC and CD154-PE, could not follow these further experiments, because we could not take up the prospective side of the study for personnel and cost reasons, this is unfortunately impossible from an organisational and capacity point of view. The intention of us was therefore purely retrospective and empirical. We hope to be able to apply for funding through the publication of these first data in order to be able to translate our plans into concrete experiments and actions.

However, despite the availability of a variety of methods and devices, there is still no test system available that could cover all aspects of platelet function which includes adhesion, aggregation, degranulation, and plug formation. Reliability, reproducibility, and robustness as well as feasibility in different laboratory and clinical environments are still the great challenges in platelet function testing.

The determination of platelet function in healthy women of reproductive age was performed to identify the decision criteria for the hyperaggregability of platelets. Thus, in this study, the dose-dependent aggregation of platelets in patients was retrospectively assessed by comparing platelet function from healthy controls.

4). For the platelet aggregation assays, the authors did not calibrate the concentrations of platelets in PRP among patients. It can be a potential drawback, as the platelet concentrations in SPS patients are almost 30% higher than healthy control. Typically, the same concentration of platelets in PRP should be used when comparing between samples. However, the PRP of SPS patients still exhibit a dose-dependent aggregation in response to ADP, thus acting as a “negative control” for their epinephrine phenotype. The authors should include this kind of discussion in the manuscript.

Response:

We sincerely thank the referee for this very careful attention. We have now provided additional information in the experimental section (see page 5, lines 188-193) and discussed this issue in the discussion section (see page 12, lines 452-462).

Platelet function for diagnostic purposes in patients at risk of thrombosis is poorly standardized. Preanalytics and sample preparation for LTA are therefore of particular importance. In this context, there are various recommendations for the preparation and possible adjustment of the PRP. Within the scope of the present work, the preparation of PRP and PPP were performed according to the recommendations of the Medical Standards Committee of the German Institute for Standardization (Deutsches Institut für Normung e.V., DIN) for the performance of a platelet function test (DIN SPEC 58961: 2013-04). The respective DIN standard is one of the most detailed descriptions of the pre-analytical procedure for performing platelet stimulation testing. According to this, an adjustment of the PRP platelet count is not recommended, although the literature discusses this fact controversially. We have taken up and discussed this fact in the discussion. Thus, CLSI guidelines give laboratories the option of testing LTA using native or platelet count-adjusted PRP samples (CLSI. Platelet Function Testing by Aggregometry: Approved Guideline. CLSI document H58-A. Wayne, PA: Clinical and Laboratory Standards Institute; 2008).

In our study, we did not adjust the platelet count of PRP during the assessment of PRP quality. One additional reason is the fact that in all samples the platelet counts in PRP samples were not lower than 150 x 109/L. However, according to Cattaneo M. et al. 2003 the platelet count of PRP samples should not be adjusted to a standardized value with autologous PPP. Thus, recent studies demonstrated that platelet counts in PRP within the range that is observed in PRP samples from subjects with normal platelet count in whole blood do not affect the results of LTA studies. Therefore, the common practice of adjusting the platelet count in PRP with autologous PPP is not recommended, because it is unnecessary and may impair the platelet responsiveness to agonists. Uncertainty remains over what is the best practice to follow when the platelet count in PRP exceeds about 600 x 109 /L. Abnormalities of platelet aggregation were more frequent using adjusted platelet count both in controls and patients (Cattaneo M. et al. J Thromb Haemost. 2009).

According to Breddin (Platelets, 2005) the PRP and PPP mixing process during platelet count adjustment in PRP leads to some platelet activation and that it should be kept in mind that the aggregation response does not change much between platelet counts of 150 x 109/L and 450 x 109/L.

In the overall view, it must therefore be stated that there is still controversy concerning whether the platelet count should be adjusted or not (Linnemann B. et al. J Thromb Haemost. 2008). On the one hand, it has been argued that in vitro aggregation is basically influenced by the platelet count in PRP and thus platelet count adjustment is recommended. On the other hand, PPP may contain substances affecting platelet function that are released by platelets or other blood cells during centrifugation of blood samples, which is necessary to obtain PPP (Breddin H.K. Platelets. 2005) (see page 12; lines 452-462)

In particular, as the reviewer mentioned correctly, normal pregnancy is characterized by a decrease in the number of circulating platelets with gestation. A possible explanation for the nevertheless possible, but not significant increase in the median platelet count in patients compared to the healthy volunteers could be (a) reactive, (b) lack of pregnancy-related complications, such as preeclampsia and (c) the various preexisting diseases of the patients listed in Table 2 - in particular internal, endocrine and gynecological disorder.

5). The authors should discuss why these SPS patients only exhibited the hypersensitivity to epinephrine, not ADP, in their platelet aggregation assays. What is the potential genetic basis behind this?

Response:

We thank the reviewer for addressing this important aspect and have provided additional information in the discussion section (see page 10, lines 347-362). We agree that the genetic background of SPS needs to be revealed as it is not known for sure at this time. We have taken up this issue in the discussion and deepened it.

In our study, type 1 (platelet hyperaggregability after both ADP and epinephrine) and type 3 (platelet hyperaggregability after ADP only) SPS were not found. According to Kubisz et al. (Ref. 35) type 2 (platelet hyperaggregability after epinephrine only) is most common, followed by type 1, whereas type 3 is rare. However, it is important that this classification is based on laboratory characteristics and no relation or differences to the clinical manifestation, treatment, or prognosis of patients were seen among the types so far. Interestingly, type 2 is the most frequent variant of SPS in white populations, whereas type 1 is the most frequent variant in Mexican mestizos (Ref. 35). The results of several genetic analyses suggested the link between certain single nucleotide polymorphism (SNPs) of membrane platelet glycoproteins. The identification of genetic changes of GPIIIa, Gas6 protein and GP6 as factors influencing platelet aggregation were promising targets in the search for the cause of SPS. However, despite the clear clinical definition and strong evidence of familiar occurrence, published results so far failed to identify a single genetic defect responsible for SPS. In sum, the laboratory heterogeneity of the syndrome with three clearly distinct types might suggest that SPS might have a multifactorial genetics. Furthermore, because SPS is made solely on the clinical and laboratory criteria and not on genetic testing, inherited and acquired changes of platelet aggregation may be included in the currently diagnosed patients. It is therefore also likely that the SPS itself does not cause thromboembolic events, similar to protein C resistance or protein C deficiency, but only predisposes to them. It is suspected that an additional factor is required for the development of a clinically relevant thromboembolic event (Bick R.L. Hematol Oncol Clin North Am. 2000). Which additional factors play the most prominent role was not identified. In our study, many secondary diagnoses of the investigated patients were identified and thromboembolic risk factors such as smoking or overweight were analyzed as prognostic factors for SPS. However, none of the secondary diagnoses showed prognostic significance for the concomitant presence of SPS. Therefore, it seems important for future investigations to identify the additional factors that might contribute to the occurrence of a thromboembolic event in patients diagnosed with SPS in order to consider these factors individually for further studies.

Minor comments:

1) The authors should cite some papers from animal studies (e.g.  Li et al, JCI, 2011, PMID: 22019589), which directly demonstrated the causal relationship between platelet hyperaggregability and pregnancy loss.

Response:

We sincerely thank the referee for this suggestion and have provided additional information based on the publication by Li C et al. in the introduction section (see page 2; lines 69-70).

2) The authors should do a better proof-reading, to avoid the typos in the paper (e.g. life births in lines 35 and 292)

Response:

We thank the reviewer for his very careful attention. We sincerely apologize for this error and corrected this typo. 

3) The subtitle of “Dose-independent platelet aggregation” (line 244) is inappropriate, as the SPS patients still exhibit dose-dependent aggregation in response to ADP.

Response:

We appreciate this comment and have added the requested correction about the subtitle in the revised manuscript: Dose-independent Epinephrine Induced Platelet Aggregation (see page 6; line 253).

Round 2

Reviewer 1 Report

Most questions were properly addressed. I have no other question in current draft. 

Author Response

Response to Reviewer 1 Comments

Round 3

Point-by-point response to the reviewer:

Thank you very much for the thorough and fair review of our manuscript.

 Reviewer:

Most questions were properly addressed. I have no other question in current draft. 

Response:

We thank the reviewer once again for the previous, very helpful comments and hope that our manuscript will now be acceptable for publication.

Reviewer 2 Report

Even taking into account that it is a retrospective study, the manuscript lacks some important scientific assessments necessary to support the finding of the study.

Author Response

Response to Reviewer 2 Comments

Round 2

Point-by-point response to the reviewer:

Thank you very much once again for the thorough and fair review of our manuscript.

Reviewer:

Even taking into account that it is a retrospective study, the manuscript lacks some important scientific assessments necessary to support the finding of the study.

Response:

We thank the referee for his/her thoughtful comments. As already mentioned previously, the analyses of homocysteine metabolism, soluble P-selectin, E-selectin or pentraxin 3, markers for platelet activation (CD62, CD41, PAC1 and CD154) or plasminogen activator inhibitor (PAI)-1 4G/5G insertion-deletion mutations and coagulation factor XIII Val34Leu polymorphism were not performed at the time of data collection. We would like to stress the fact that these complex analyses are rarely performed together with platelet function tests in daily routine, in particular because more specialized analyses are reserved for highly specialized laboratories and the platelet function tests for experienced laboratories. Due to the retrospective evaluation of patient data over a long period of time and from the routine clinical work up, respectively, a further analytical approach with blood samples from patients and healthy controls is not applicable.

To address your remaining concern, we have critically highlighted that these aspects need to be taken into account in further studies (see page 12; lines 468-475).

Reviewer 3 Report

The Reviewer appreciated the detailed point-by-point response from the Authors. The Authors should indicate "the explorative nature of the work" in the manuscript, and include "analysis of more SPS patients without ASA" as a future direction. 

Also in the Discussion, the Authors should discuss about the potential contribution of plasma to the current LTA assays, and include the analysis of gel-filtered platelets or/and plasma switch in LTA as a future direction.

If the Authors couldn't complete the flow cytometry assays to analyze the platelet activation, at least they should include them as a future direction in the manuscript.

Author Response

Response to Reviewer 3 Comments

Round 2

Point-by-point response to the reviewer:

Thank you very much for the thorough and fair review of our manuscript.

Reviewer:

The Reviewer appreciated the detailed point-by-point response from the Authors. The Authors should indicate "the explorative nature of the work" in the manuscript, and include "analysis of more SPS patients without ASA" as a future direction. 

Response:

We sincerely thank the referee for this suggestion and have provided additional information in the introduction (see page 3; line 103) and discussion section (see page 10; lines 366-371). Due to the low number of patients both without ASA therapy but successful pregnancy, the diagnostic and clinical relevance of less successful pregnancy in these patients might be premature at this stage of this explorative statistical analysis. However, further research should aim at investigating more SPS patients without ASA in regard to pregnancy outcome and to clarify the exact pathogenic and therapeutic role of platelet hyperaggregability in clinical setting.  

Also in the Discussion, the Authors should discuss about the potential contribution of plasma to the current LTA assays, and include the analysis of gel-filtered platelets or/and plasma switch in LTA as a future direction.

Response:

We appreciate this comment and have provided additional information in the discussion section. As we have previously discussed this issue in the discussion section (see page 11-12; lines 441-455) citrated plasma is mainly used to diagnose platelet function. However, citrate plasma is not the optimal medium for platelet function testing. Therefore, further studies should be conducted to investigate the diagnostic value of other high quality analytical strategies based on gel-filtered platelets or/and plasma switch in LTA analysis (see page 12; lines 466-468).

If the Authors couldn't complete the flow cytometry assays to analyze the platelet activation, at least they should include them as a future direction in the manuscript.

Response:

We thank the reviewer for his/her comment and have provided additional information in the discussion section. Flow cytometry for detection of platelet-specific activation markers, such as markers for platelet activation (P-Selectin and fibrinogen binding or others such as CD62, CD41, PAC1 and CD154) may potentially be explored in further research of platelet function as diagnostic approach potentially providing more objectivity and stability of results (see page 12; lines 468-475).

This manuscript is a resubmission of an earlier submission. The following is a list of the peer review reports and author responses from that submission.

Round 1

Reviewer 1 Report

Yagmur et al., described their findings in correlation between sticky platelet syndrome(SPS) with infertility and pregnancy loss. This is a hospital based retrospective study and results are interesting.

1. Similar findings have been described in several papers and review articles. The authors should try to point out their novelty.

2. The data presentation is not proper, some descriptions in the Result Section indicated “data not shown”. It is not clear that why the authors described their findings under some analyses but no showing the analytic data.

3. Infertility and pregnancy loss should belong to different criteria. Here, the authors just mentioned the potential correlation of SPS with pregnancy loss in the introduction. This should be revised. Also the results combined the factors involved in infertility and pregnancy loss. These correlated factors should be introduced in the introduction.

4. In table 1, it was described “29 age-matched healthy, non-pregnant, fertile women as a diagnostic control group”, it seems not consistent with the results and legends.

5. The authors need explain the reasons that why the data in Table 1 showed no significance of SPS with pregnancy loss and all factors correlated with infertility and pregnancy loss.   

6. The rationale of SPS linking with ASA Therapy in the Results should be explained. Some background of ASA therapy should introduced in the Introduction.